# Examining Criteria for Choosing Subcontractors for Complex and Multi-Systems Projects

**Shimon Fridkin [1] and Sigal Kordova [2,*]**

1    Faculty of Industrial Engineering and Technology Management, HIT—Holon Institute of Technology, Holon 5810201, Israel
2    Department of Industrial Engineering and Management, Ariel University, Ariel 40700, Israel
*    Correspondence: sigalko@ariel.ac.il or sigalkord@gmail.com

**Abstract:** Numerous companies from diverse industries use subcontracting in their operations. In complex projects, subcontractor selection is a crucial managerial decision that significantly impacts project success. The current mixed-methodology study examines that criteria that high-tech defense and civilian companies use to choose optimal subcontractors. The qualitative aspect derives from semi-structured interviews; the quantitative findings were obtained using three statistical methods: Friedman's two-way analysis of variance by ranks, hierarchical cluster analysis, and multidimensional scaling (PROXSCAL). Data analysis yielded twelve leading criteria for subcontractor selection, categorized into four clusters of varying strength. The three highest-rated criteria were significantly stronger than the others and included system reliability and quality, level of service, and flexibility to change. The lowest rated criteria were leadership and innovation, and number of systems supplied in the past. The findings provide practical insights applicable to subcontractor selection and expand our knowledge of complex project management.

**Keywords:** subcontractors; complex projects; project management; hierarchical cluster analysis

## 1. Introduction

Subcontracting is an operational practice of many companies in varied industries. A subcontractor employed by a company answers to that company only, although it is not the work's end customer [1]. Once the abilities and needs required for a project are defined, external agents may provide them either as a one-time collaboration in a specific task within a limited timeframe or as a long-term working relationship. Subcontracting allows companies to focus on core technologies and business which are critical to their success, rather than investing in resources, infrastructure, and knowledge-creation that are not as relevant for their core activity, thereby reducing costs and saving time. Choosing a subcontractor is a crucial managerial decision that significantly impacts the success of complex projects. A poor selection can have serious ramifications, leading to considerable financial losses, delays, and even project failure.

A review of the academic literature reveals that few studies analyze subcontractor selection or develop a model for the selection process. Most of the previous studies engage with construction companies [2–4]. Since no criteria have been determined yet for subcontractor selection in other domains, addressing this issue is required further the previous studies.

The goal of this study is, therefore, to discuss criteria used in choosing subcontractors for complex, multi-system projects, particularly ones that contribute the most to choosing an optimal subcontractor. Indeed, the success of any project largely depends on selecting the best possible subcontractors, capable of reliably supporting the company's capability to deliver the project in compliance with the quality, schedule, and cost requirements [5].

This study examines subcontractor selection criteria used by defense and civilian high-tech industries. We expect these criteria to form a spectrum ranging from highest to minor importance. The research design maps this spectrum.

Our research question is, "What are the main criteria for subcontractor selection in complex, multi-system high-tech projects?"

## 2. Literature Review

### 2.1. Subcontracting

According to The Project Management Institute, "subcontracting occurs when the primary contractor assigns, leases, outsources, or employs another person to support the primary contractor in executing part of a project, according to the terms of a contract" [6]. Thus, a subcontractor is an individual or a company who takes responsibility for carrying out work for another producer, who has a larger work contract (the *prime* or *main contractor*). A subcontractor is sought when the main contractor cannot carry out the work unaided, either because of a lack of time or because employing a subcontractor is less expensive than doing the work in house [5].

It is critical that the main contractor decides wisely when planning how to execute a project. Once a company opts to involve a subcontractor, the main contractor must be fully aware that it has now accepted an additional responsibility, that of managing the subcontractor's work. It is important to ensure that the main contractor has the capacity required to supervise one or more subcontractors, because various parts or stages of the project necessitate employing different subcontractors [7].

As the world has become an accessible global village, the use of subcontractors has increased because the number of suppliers capable of offering quality technological products has also grown. This expansion has, in turn, intensified competition, so that many technology companies prefer to work only with their own core technology and obtain everything else from subcontractors [5].

In other cases, a subcontractor sells its own core technologies developed according to the main contractor's needs. Thus, subcontractors can help a company realize and implement its original idea. Due to globalization, the number of subcontractors involved in a project can sometimes be such that a significant part of a product is manufactured outside its country of origin [8]. Subcontracting is a common practice in the construction industry [9]. The success of a general construction contractor depends primarily on subcontractors' performance [10]. Therefore, selecting the right subcontractor for each portion of a job is critical not only for the successful completion of the particular project but also for the reputation and business continuity of the general contractor [5,11–14].

Several earlier studies have focused on identifying and rating the importance of subcontractor selection criteria [3,5,11–13,15]. In the construction industry, the predominant criteria for selecting subcontractors are price, reputation, techniques, ability to complete projects as required, commitment, and subcontractor competence [4]. Other studies propose tools, techniques, and/or methodologies for subcontractor selection, e.g., using a blockchain-enabled smart contract as trust-enhancing technology [12,16–18].

Some of the mentioned studies focus on selecting the most appropriate subcontractor for a specific work package within a project. Others present the subcontractors' perspectives and their impact on project success [19–36]. Stadnicka and Ratnayake suggest using a tailor-made value stream mapping (VSM) to assess the challenges and mitigate the delays related to preparing the quotation for selecting subcontracting manufacturers [37].

Other than managing the subcontractors' role in the technical and specific aspects of a project, the success of a project also requires building trust with them. Trust arises over a psychological dialogue between people and groups. Once the sides create positive shared experiences, they will be able to form a partnership that contributes to the success of the project's success, especially when subcontractors execute a significant share of the overall scope [38].

Manu et al. listed several factors that influence the creation of trust between main contractors and subcontractors [39]:

- Changes in the management process: A controversy about changes in the scope of work influences the level of trust between the two parties. The main contractor must be aware of unexpected changes in the scope of work, while the subcontractor must be honest regarding added costs, if any.
- Payment practices: The subcontractor must be paid on time for the work done. This will reduce the subcontractor's financial stress and increase his trust in the main contractor.
- Changes in the economic climate: Negative changes (such as a recession) generally cause project managers to prefer incurring costs (at least in the short-term) rather than become dependent on quality subcontractors who are more expensive, but also provide higher quality work in the long run.
- Future business opportunities: Successful past projects and prospects of a continuous stream of sizeable projects in the future will increase a subcontractor's trust in the main contractor.
- Quality performance: If problems arise, the main contractor expects transparency from subcontractors but also expects them to know how to respond and solve problems by hiring stable high-quality workers.
- Specific project-related circumstances: Hiring new subcontractors, whether local or foreign, is often accompanied by significant concern about their suitability for the task, especially in large, complex projects that require subcontractors with specific skills. The trust built with subcontractors will help determine the project's success [39]. Akal emphasizes that all tender documents relevant to a subcontract must be intelligible and written so as to ensure that both parties interpret their terms in the same way [40].

None of the earlier studies discuss the need to examine the criteria for subcontractor selection in the dynamic and unpredicted reality of complex, multi-system projects. The next section of the literature review describes the unique characteristics of a complex project and their impact on the project's outcomes.

*2.2. Complex Projects*

The current study engages with high-tech projects that require multi-disciplinary knowledge to develop new products, systems, and processes. According to the ICCPM definition, a complex project is one with high levels of uncertainty, irregularity, emergence, dynamic complexity, instability, and constantly changing interactions between its many components [41]. Complex as they already begin, complexity continues to increase as the projects progress.

Earlier studies have analyzed project complexity through the prisms of their organizational, cultural, environmental, technological, information, and goal complexity [41–43]. Complexity impacts outcomes, completion, and success, sometimes interacting with other factors relevant to the project [42,44–46]. Previous studies that have focused on the effect of complexity and interaction with managerial strategies on project outcomes have yielded mixed results. For example, one study found that stakeholder engagement has a negative effect on quantitative project performance but a positive effect on qualitative project performance [46]. The project and the system it delivers, be it a product, a service, or some combination thereof, are inseparable. The system's architecture, requirements, and design determine the components to be developed, tested, and delivered [47].

According to SEBoK (2022), a complex system (CxS) is a project that combines a core technology and at least two additional engineering fields [48]. The core technology must be the organization's main focus, e.g., inspection, vision, algorithms, and printing. The engineering fields may include mechanics, electronics, programming, optics, pneumatics, or others. Rouse claims that complex systems include a large number of independent heterogeneous components, with many and varied characteristics, and the system's boundaries are often difficult to determine [49]. Interactions between the various components

have dramatic impact on the final outcome/product and largely determine the complex system's structure and behavior [50].

Since the criteria of subcontractor selection for complex projects have not been reported in the literature review, the present study analyzes these criteria.

## 3. Methodology

To improve our understanding of the best way to select subcontractors for complex projects, we used combined qualitative and quantitative methods, as described below. The quantitative aspect leaned on one classical method (Friedman's two-way analysis of variance by ranks; hereinafter, "Friedman's Two-way Analysis") and two additional advanced statistical methods: hierarchical cluster analysis and multidimensional scaling (PROXSCAL). Combining these three methods is required in the current study, given the relatively small convenience sample of 7 respondents used to examine twelve criteria. The study stands at the seam between a quantitative and a qualitative approach. Friedman's two-way analysis allows statistical inference from a sample population (i.e., a quantitative approach) while hierarchical cluster analysis and PROXSCAL facilitate a qualitative approach, in which the sample is viewed as a population, a kind of case study. In PROXSCAL, the goodness of fit indicates the uniqueness of the findings. Combining the two methods allowed us to develop an integrative view in classifying the criteria for subcontractor selection. Combining the three mentioned methods has an advantage over other statistical methods, such as a fuzzy analytic network process analysis [43], since these methods are sufficient and do not require further validation using additional statistical methods.

We conducted the study in two stages as presented in Figure 1: a first qualitative procedure followed by a quantitative one. Initially, we conducted semi-structured interviews with six senior academic and industry experts. All of the interviewees had a rich background and broad experience in engineering systems, purchasing management, and working with subcontractors. The interviews' purpose was to identify the various criteria used in choosing subcontractors for complex, multi-system projects. The interviewees had to answer several questions regarding the considerations for selecting subcontractors for complex projects and the impact of these considerations on the project outcome. Qualitative content analysis was used to analyze the interviews and find common criteria that interviewees used for choosing subcontractors. After sorting the data extracted from the interviews into categories and themes, an analysis followed of the prevalence and incidence of repetitions. The criteria that the interviewees mentioned most frequently received special attention. To establish the trustworthiness of the qualitative study, we analyzed all the experts' insights collectively.

Furthermore, we used triangulation to validate the categories and implemented cross-content analysis to examine their interrelations.

Following the content analysis and triangulation, we chose the 1 most frequently encountered categories as the criteria for the questionnaire of stage 2.

These 1 criteria served to put together the first version of a questionnaire. Before distributing the questionnaire, we asked five leading experts in systems engineering, project management, complex projects, and managing sub-contractors in defense and civilian industries to validate its first version, rate the relevance and clarity of each item on a scale of 1 (very low) to 5 (very high), and suggest changes and corrections.

Based on this expert feedback, we calculated the Fleiss Kappa index to evaluate the degree of agreement among the experts regarding the relevance and clarity of the questionnaire items. The Fleiss kappa index result was 0.78, indicating a significant agreement between the experts. Several changes and updates were made to the phrasing of the questionnaire's first version following the experts' feedback.

Stage 1:

**Qualitative Study**

---

**Semi-structured Interviews:**

- Content Analysis

- Triangulation

---

Stage 2:

**Quantitative Study**

---

First Version of the

Questionnaire

---

Validation by Experts

---

Final Version of the

Questionnaire

---

Data Collection

---

Data Analysis

---

Results

**Figure 1.** The study design.

We distributed the final version of the questionnaire via Google Forms to 12 engineers and other senior executives of civilian and military high-tech industries, and 7 of them filled the questionnaire. The research population consisted of professional employees with recognized engineering and management skills from the civilian ($n = 45$) and defense ($n = 31$) industries, who had been previously involved in projects of different sizes. The industries varied from large, multinational companies to small companies involved in developing and marketing specific technologies. The respondents included systems engineers, developers, purchasing managers, and operations managers. All were directly involved in or connected with choosing subcontractors.

The first section of the questionnaire gathered information about the respondent: age, education, general and specific professional experience, the industry in which they work, their position, and characteristics of their most recent project that required hiring

a subcontractor. Concerning the project, they were asked the project type, its level of complexity, and the financial scope of the portion in which subcontractors were employed.

The second, principal section of the questionnaire presented 1 leading criteria for choosing subcontractors, which were derived from both the interviews and professional literature. The respondents were asked to rank the contribution of each criterion separately, on a scale from to 10. The criteria included were:

- Criterion 01: Experience in the field;
- Criterion 02: Leadership and innovation;
- Criterion 03: Flexibility to changes;
- Criterion 04: Previous work relationships;
- Criterion 05: Pricing/costs;
- Criterion 06: Level of service;
- Criterion 07: Quality standards and procedures;
- Criterion 08: Complements missing abilities;
- Criterion 09: Availability;
- Criterion 10: Number of systems supplied in the past;
- Criterion 11: Technological maturity;
- Criterion 12: System reliability and quality.

Reliability was examined by checking for internal consistency using the questionnaire's alpha Cronbach reliability. The result was 0.83, which reflects the questionnaire's high internal reliability.

*Statistical Analyses*

Responses to the questionnaire were analyzed with IBM SPSS 27.0 [51]. Before testing the research questions, we calculated the distribution of the 1 selected criteria. In the current study, the statistical analysis leaned on three nonparametric statistical methods: a related-samples Friedman's two-way analysis of variance by ranks, hierarchical cluster analysis, and multidimensional scaling (PROXSCAL). Ranks are unique in that they represent the ordering of a numeric variable's value. Being the cornerstone of many nonparametric statistical methods, they are useful in computing the rank transform of a variable in a dataset. Because ranks are the cornerstone of many nonparametric statistical methods, it is useful to compute the rank transform of a variable in a dataset. To answer the research question, we used a related-samples Friedman's two-way analysis of variance by ranks (hereinafter, "Friedman's Two-way Analysis), including pairwise comparisons analysis. As a nonparametric test, Friedman's two-way analysis is used with ranked data, particularly for cases where the data do not meet the rigor of interval data, and when there are serious concerns about extreme deviation from normal distribution. Because this test does not assume that data has a particular distribution (e.g., normal distribution), it is used in place of the ANOVA test when the distribution of data is unknown. Moreover, Friedman's two-way analysis is an extension of the sign test for situations with multiple treatments. In fact, if there are only two treatments, the tests are identical.

Those findings were then validated using hierarchical cluster analysis, which is helpful for identifying relatively homogeneous groups of cases (or variables) based on selected characteristics. It uses an algorithm that starts with each case (or variable) in a separate cluster and combines clusters until only one is left. Finally, multidimensional scaling (PROXSCAL) was used to further validate the findings of the first two statistical tests. PROXSCAL is designed to reveal the structure in a set of proximity measures between objects, by assigning observations to specific locations in a low-dimension, conceptual space, such that the distances between points in the space match the given (dis)similarities as closely as possible. The result is a least-squares representation of the objects in that low-dimensional space, which, in many cases, will help to further understand data.

## 4. Results

Findings of the Kolmogorov–Smirnov test for normality provided evidence that the distribution of the 1 criteria for selecting subcontractors is abnormal. Table 1 shows descriptive statistics for the criteria, including mean rank, mean, standard deviation, measurement scale, type of distribution, fit statistics, and distribution parameters.

Our research question asked: What are the main criteria for choosing subcontractors for complex, multi-system projects? Figure 2 shows the results of the Friedman's two-way analysis conducted to answer that question.

**Table 1.** Descriptive statistics of 1 subcontractor selection criteria: mean rank, mean, standard deviation, measurement scale, type of distribution, fit statistics, and distribution parameters (*n* = 76).

| Criterion 01: Experience in the Field | | Criterion 02: Leadership and Innovation | |
|---|---|---|---|
| Mean Rank | 7.38 | Mean Rank | 4.72 |
| M(SD) | 8.45 (1.27) | M(SD) | 7.37 (1.52) |
| Treat as Ordinal: | | Treat as Ordinal: | |
| Type of Distribution | Binomial | Type of Distribution | Binomial |
| Fit Statistics | $\chi^2 = 41.00$ *** | Fit Statistics | $\chi^2 = 7.40$ |
| Parameters | n = 10; prob = 0.84 | Parameters | n = 11; prob = 0.67 |
| Treat as Continuous: | | Treat as Continuous: | |
| Type of Distribution | Triangular | Type of Distribution | Triangular |
| Fit Statistics | A = −2.24 ***; K = 0.34 *** | Fit Statistics | A = 0.61 ***; K = 0.20 *** |
| Parameters | max = 10; min = 4; mode = 10 | Parameters | max = 10; min = 4 mode = 8.11 |
| **Criterion 03: Flexibility Concerning Changes** | | **Criterion 04: Previous Work Relationships** | |
| Mean Rank | 7.77 | Mean Rank | 5.79 |
| M(SD) | 8.57 (1.25) | M(SD) | 7.68 (1.86) |
| Treat as Ordinal: | | Treat as Ordinal: | |
| Type of Distribution | Binomial | Type of Distribution | Poisson |
| Fit Statistics | $\chi^2 = 28.87$ *** | Fit Statistics | $\chi^2 = 30.57$ *** |
| Parameters | n = 10; prob = 0.86 | Parameters | Mean = 7.68 |
| Treat as Continuous: | | Treat as Continuous: | |
| Type of Distribution | Triangular | Type of Distribution | Triangular |
| Fit Statistics | A = −8.43 ***; K = 0.25 *** | Fit Statistics | A = −2.34 ***; K = 0.29 *** |
| Parameters | max = 10; min = 5; mode = 10 | Parameters | max = 10; min = 1; mode = 10 |
| **Criterion 05: Pricing/Costs** | | **Criterion 06: Level of Service** | |
| Mean Rank | 5.89 | Mean Rank | 8.02 |
| M(SD) | 7.86 (1.50) | M(SD) | 8.55 (1.34) |
| Treat as Ordinal: | | Treat as Ordinal: | |
| Type of Distribution | Binomial | Type of Distribution | Binomial |
| Fit Statistics | $\chi^2 = 7.46$ | Fit Statistics | $\chi^2 = 27.31$ *** |
| Parameters | n = 11; prob = 0.71 | Parameters | n = 11; prob = 0.78 |
| Treat as Continuous: | | Treat as Continuous: | |
| Type of Distribution | Triangular | Type of Distribution | Triangular |
| Fit Statistics | A = −2.33 ***; K = 0.14 *** | Fit Statistics | A = −6.98 ***; K = 0.33 *** |
| Parameters | max = 10; min = 4; mode = 9.57 | Parameters | max = 10; min = 4; mode = 10 |

**Table 1.** *Cont.*

| Criterion 07: Quality Standards and Procedures | | Criterion 08: Complements Missing Capabilities | |
|---|---|---|---|
| Mean Rank | 7.06 | Mean Rank | 6.52 |
| M(SD) | 8.25 (1.45) | M(SD) | 8.09 (1.47) |
| Treat as Ordinal: | | Treat as Ordinal: | |
| Type of Distribution | Binomial | Type of Distribution | Binomial |
| Fit Statistics | $\chi^2 = 15.67$ | Fit Statistics | $\chi^2 = 9.54$ |
| Parameters | n = 11; prob = 0.75 | Parameters | n = 11; prob = 0.74 |
| Treat as Continuous: | | Treat as Continuous: | |
| Type of Distribution | Triangular | Type of Distribution | Triangular |
| Fit Statistics | A = −7.64 ***; K = 0.25 *** | Fit Statistics | A = −4.57 ***; K = 0.18 *** |
| Parameters | max = 10; min = 4; mode = 10 | Parameters | max = 10; min = 4; mode = 10 |
| Criterion 09: Availability | | Criterion 10: Number of Systems Supplied in the Past | |
| Mean Rank | 7.15 | Mean Rank | 3.36 |
| M(SD) | 8.37 (1.30) | M(SD) | 6.54 (1.97) |
| Treat as Ordinal: | | Treat as Ordinal: | |
| Type of Distribution | Binomial | Type of Distribution | Binomial |
| Fit Statistics | $\chi^2 = 14.43$ | Fit Statistics | $\chi^2 = 20.90$ |
| Parameters | n = 10; prob = 0.84 | Parameters | n = 16; prob = 0.41 |
| Treat as Continuous: | | Treat as Continuous: | |
| Type of Distribution | Triangular | Type of Distribution | Triangular |
| Fit Statistics | A = −7.15 ***; K = 0.18 *** | Fit Statistics | A = 1.90 ***; K = 0.21 *** |
| Parameters | max = 10; min = 5; mode = 10 | Parameters | max = 10; min = 2; mode = 7.62 |
| Criterion 11. Technological Maturity | | Criterion 12. System Reliability and Quality | |
| Mean Rank | 5.88 | Mean Rank | 8.46 |
| M(SD) | 7.91 (1.31) | M(SD) | 8.82 (1.00) |
| Treat as Ordinal: | | Treat as Ordinal: | |
| Type of Distribution | Binomial | Type of Distribution | Binomial |
| Fit Statistics | $\chi^2 = 21.40$ ** | Fit Statistics | $\chi^2 = 5.18$ |
| Parameters | n = 10; prob = 0.79 | Parameters | n = 10; prob = 0.88 |
| Treat as Continuous: | | Treat as Continuous: | |
| Type of Distribution | Weibull | Type of Distribution | Triangular |
| Fit Statistics | A = 3.60 **; K = 0.24 ** | Fit Statistics | A = −7.94 ***; K = 0.30 *** |
| Parameters | b = 8.40; b = 8.16; c = 0.00 | Parameters | max = 10; min = 5; mode = 10 |

Note: ** *p* < 0.01; *** *p* < 0.001; A = Anderson-Darling Test; K = Kolmogorov-Smirnoff Test.

Figure 2 shows that the ranking of criterion 12 (system reliability and quality) and criterion 06 (level of service) is significantly stronger than criterion 05 (pricing/costs), criterion 11 (technological maturity), criterion 04 (previous work relationships), criterion 02 (leadership and innovation), and criterion 10 (number of systems supplied in the past). The ranking of criterion 03 (flexibility to changes) is significantly stronger than criterion 04 (previous work relationships), criterion 02 (leadership and innovation), and criterion 10 (number of systems supplied in the past). Therefore, criteria 12 (system reliability and quality), 06 (level of service), and 03 (flexibility to changes) can be combined into the cluster with the greatest rating power.

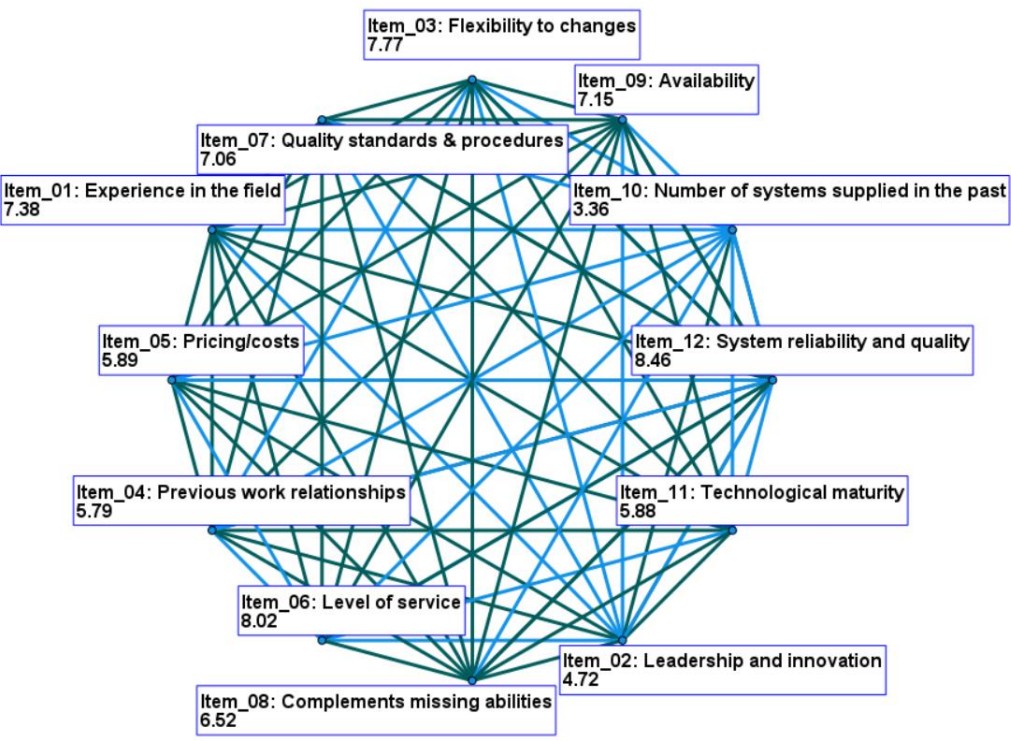

**Figure 2.** Results of related-samples Friedman's two-way analysis of variance by ranks including pairwise comparisons $\chi_F^2(11) = 166.35$, $p < 0.001$; $n = 76$; connections in blue represent significant differences; significance values have been adjusted by the Bonferroni correction for multiple tests.

The second cluster, in terms of rating strength, includes criterion 01 (experience in the field), criterion 09 (availability), and criterion 07 (quality standards and procedures). The third cluster, in terms of rating strength, includes four criteria: criterion 08 (complements missing abilities), criterion 05 (pricing/costs), criterion 11 (technological maturity), and criterion 04 (previous work relationships). Finally, criterion 02 (leadership and innovation) and criterion 10 (number of systems supplied in the past) (fourth cluster) have the weakest power of the 1 subcontractor selection criteria examined in this study.

Figure 3 shows a possible division of the 1 criteria into four groups (clusters).

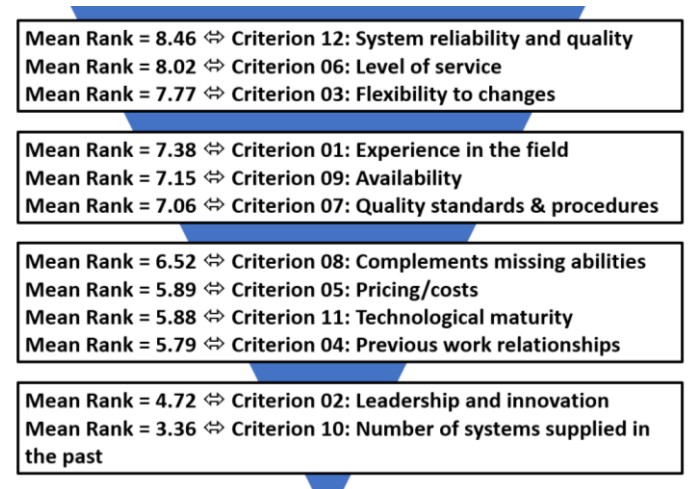

**Figure 3.** The division of the 1 subcontractor selection criteria into four groups (clusters).

To validate the insights obtained from Friedman's two-way analysis, a hierarchical cluster analysis was conducted, in which the 1 subcontractor selection criteria were used as cases, and the mean rank was used as a classification variable. Figure 4 shows the results of the analysis.

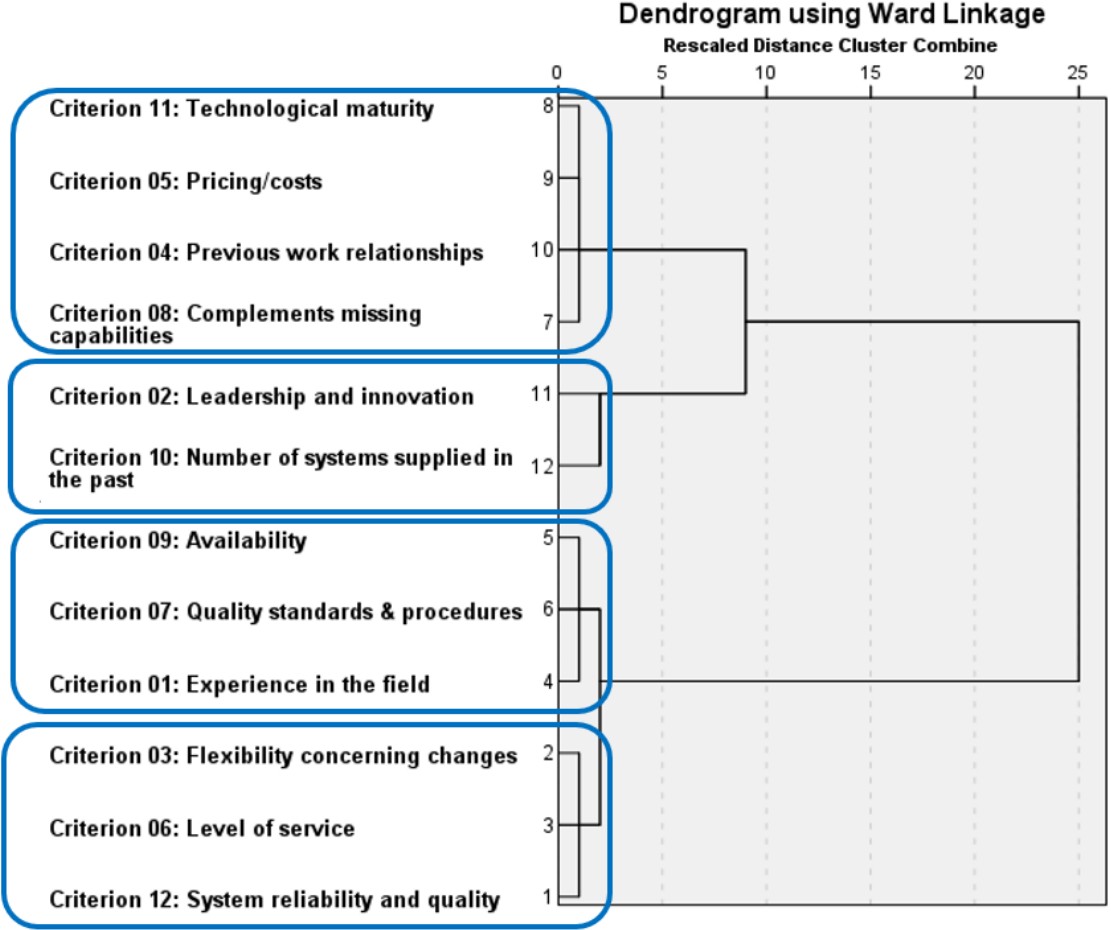

**Figure 4.** Dendrogram of hierarchical cluster analysis presenting the classification of 1 subcontractor selection criteria into four clusters (Method: Ward linkage).

The diagram shows that the results of the hierarchical cluster analysis are consistent with the results of the Friedman's two-way analysis. Criterion 12 (system reliability and quality, mean rank = 8.46), criterion 06 (level of service, mean rank = 8.02), and criterion 03 (flexibility to changes, mean rank = 7.77) constitute one cluster, criteria of very high importance (range of mean rank: 7.77–8.46). Next, criterion 01 (experience in the field, mean rank = 7.38), criterion 09 (availability, mean rank = 7.15), and criterion 07 (quality standards and procedures, mean rank = 7.06) follow in the second cluster, criteria of high importance (range of mean rank: 7.06–7.38). Criterion 08 (complements missing abilities, mean rank = 6.52), criterion 05 (pricing/costs, mean rank = 5.89), criterion 11 (technological maturity, mean rank = 5.88), and criterion 04 (previous work relationships, mean rank = 5.79) are in the third cluster, criteria of medium importance (range of mean rank: 5.79–6.52). The last cluster consists of criterion 02 (leadership and innovation, mean rank = 4.72) and criterion 10 (number of systems supplied in the past, mean rank = 3.36), which are of low importance (range of mean rank: 3.36–4.72).

To further validate the above findings, a two-dimensional PROXSCAL analysis was used to reduce the dimensions of the 1 subcontractor selection criteria. The criteria were entered into the analysis as column objects and the 7 research participants were considered as row objects. Figure 5 shows the results.

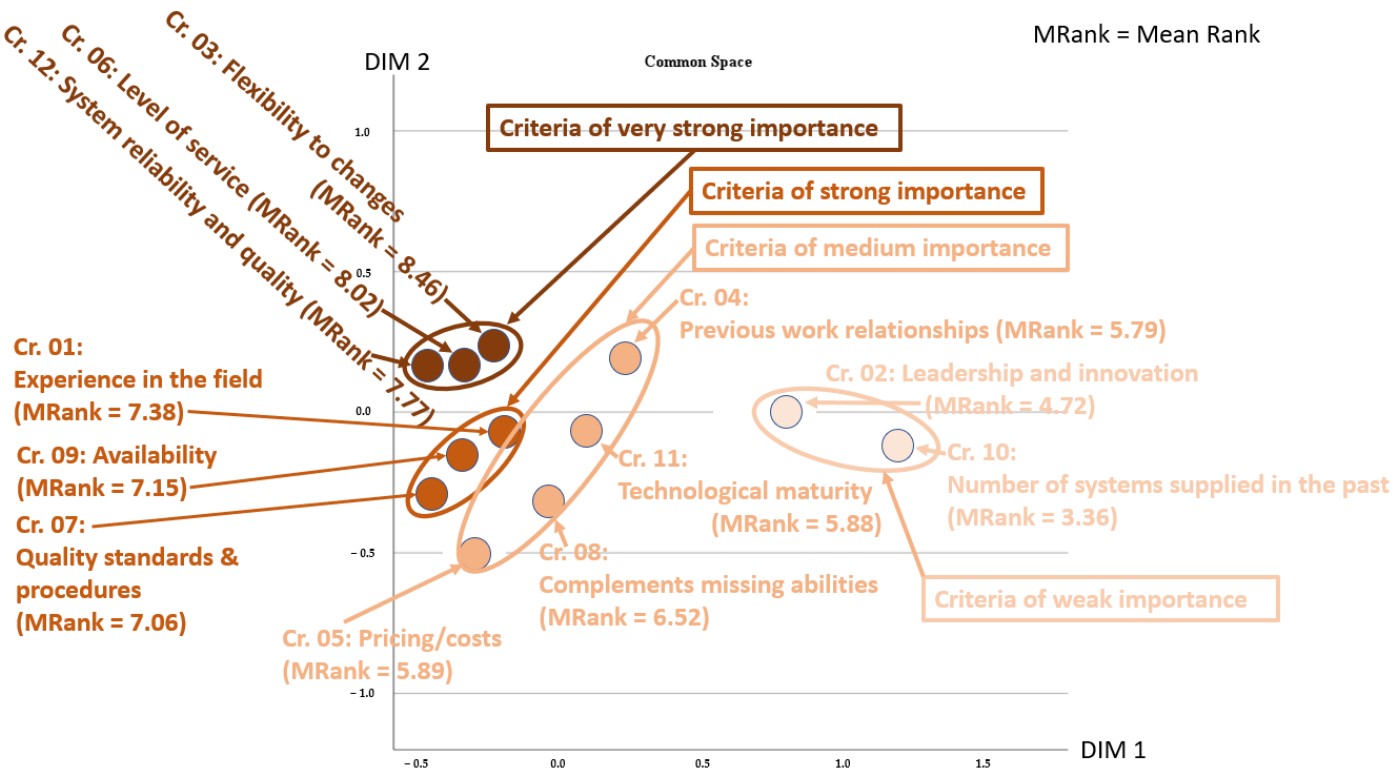

**Figure 5.** Results of two-dimensional scaling analysis (PROXSCAL) for reducing the dimensions of the 1 subcontractor selection criteria.

Two-dimensional scaling analysis yielded excellent stress and fit measures indicating a clear separation between groups of criteria: normalized raw stress = 0.00967, stress-i = 0.098338 [optimal scaling factor = 1.010], stress-ii = 0.177445 [optimal scaling factor = 1.010], s-stress = 0.013342 [optimal scaling factor = 1.001], dispersion accounted for (d.a.f.) = 0.990330, and Tucker's coefficient of congruence = 0.995153. These findings are evident in Figure 5. First, we see that criterion 12 (system reliability and quality), criterion 06 (level of service), and criterion 03 (flexibility to changes) group together into a cluster of the highest importance, with mean ranks in the range of 7.77–8.46. Next, criterion 01 (experience in the field), criterion 09 (availability), and criterion 07 (quality standards and procedures) group together into a cluster of criteria of high importance, with mean ranks in the range of 7.06–7.38. Another group of criteria that are relatively close includes criterion 08 (complements missing abilities), criterion 05 (pricing/costs), criterion 11 (technological maturity), and criterion 04 (previous work relationships), which are of medium importance, with mean ranks in the range of 5.79–5.52. Finally, a final group of the least important criteria includes criterion 02 (leadership and innovation) and criterion 10 (number of systems supplied in the past) with mean ranks in the range of 3.36–4.72. Thus, the results of the two-dimensional scaling analysis (PROXSCAL) fit the results of Friedman's two-way analysis and the hierarchical cluster analysis.

## 5. Discussion

The current study examined the principal criteria used in choosing subcontractors for complex multi-system projects. Recent publications discussing subcontracting have mainly focused on the construction industry and concluded that contractors tended to acknowledge that tender prices were the most significant selection criterion. However, relying on the "lowest bid" can potentially lead to choosing an inappropriate subcontractor, entailing damage to the company's reputation and business continuity [52].

Defining the main criteria for choosing subcontractors in complex projects is far more complicated. Complex multi-systems projects have extended timeframes and require

coordinating between multiple interdependent project stakeholders [53]. Delivering a sophisticated product, service, or technology that meets the customer's specifications requires the collaboration of suppliers and other partners. Therefore, identifying, analyzing, and prioritizing the most crucial criteria for selecting appropriate subcontractors is critical. However, since so many varied firms, teams, and individuals are involved, the criteria might prove ambiguous.

In this mixed-method study, we used quantitative and qualitative analytical tools to develop a reliable process for understanding the criteria for subcontractor selection in complex projects. The semi-structured interviews with professionals in the field yielded more than 4 possible criteria for subcontractors. Triangulation enabled narrowing the list to 1 main criteria. The respondents of a quantitative questionnaire ranked these criteria based on their experience choosing subcontractors for complex and multi-system projects.

The research findings, validated using the three described statistical methods, yielded evidence that the 1 subcontractor selection criteria split into four clusters of differing strength. The criteria whose ratings were by far the highest are system reliability and quality, quality of service, and flexibility to change. Next is a group of weaker criteria that are nevertheless significantly stronger than the third group, including experience in the field, availability, and quality standards and procedures. The third group includes complements to missing abilities, pricing/costs, technological maturity, and previous work relationships. The remaining two criteria that received the lowest rating were leadership and innovation and number of systems supplied in the past. Figure 6 summarizes the findings of the quantitative study.

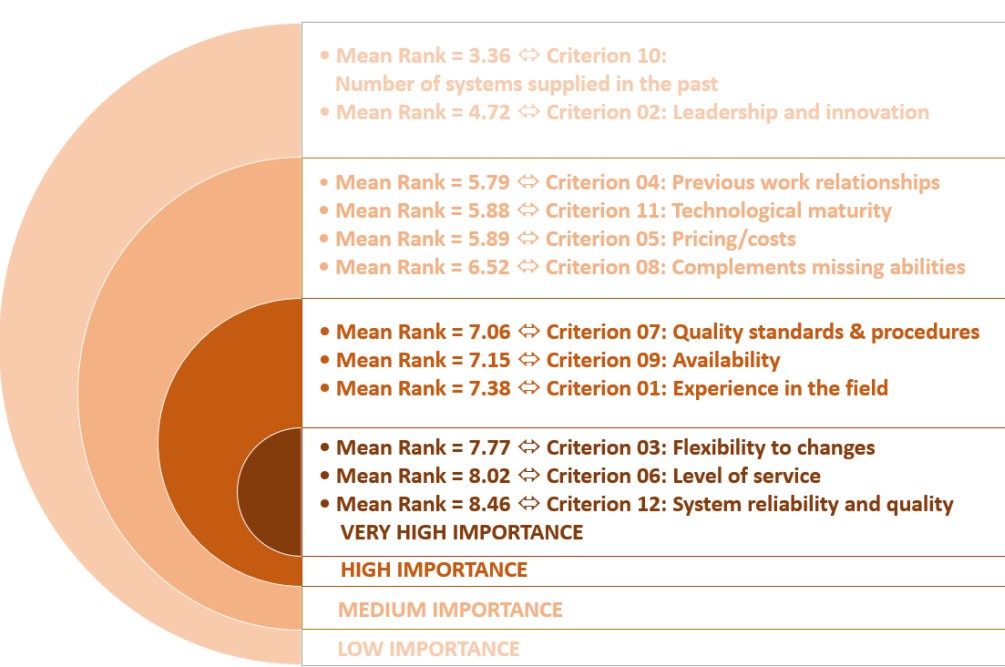

**Figure 6.** The 1 main criteria for subcontractor selection, displayed by importance.

Figure 6 illustrates the integrative evidence obtained in the study: the criteria can be treated as a spherically structured content world with heat-like characteristics. The inner circle represents the criteria rated as having the highest importance (criterion 12: system reliability and quality, criterion 06: level of service, and criterion 03: flexibility to changes), with a mean importance range of 7.77–8.46. In the next circle, we find the second-highest importance criteria (criterion 01: experience in the field, criterion 09: availability, and criterion 07: quality standards and procedures) with a mean importance range of 7.06–7.38. The third circle represents criteria of medium importance (criterion 08: complements missing abilities, criterion 05: pricing/costs, criterion 11: technological maturity, and criterion 04: previous work relationships) with a mean importance range of 5.79–6.52.

Finally, the outer circle comprises the lowest-importance criteria (criterion 02: leadership and innovation and criterion 10: number of systems supplied in the past) with a mean importance range of 3.36–4.72.

Despite examining different industries, our findings are partly consistent with those reported in the literature, particularly the indication that the potential subcontractors' reputation and the quality of their most recent projects are assigned critical importance in the prequalification and subcontractor selection process [54]. Yet, recent studies have shown that the significant factors for selecting subcontractors in the construction industry varied broadly, and included cost, quality, schedule, reputation, technical qualifications, safety, technology level, and financial status [4,55].

The current study expands the discussion of selecting subcontractors for complex high-tech projects. The characteristics and design of defense and civilian high-tech projects are usually different from those of construction projects, given their sophistication and typical multi-system integration (e.g., cyber-physical systems and complex networks). Therefore, their management follows different criteria. Analysis of the qualitative data yielded a long list of criteria used to choose subcontractors in such projects, in addition to managerial viewpoints deriving from other comprehensive variables such as project complexity, scope, and cost. Optimization models enable determining minimum project time and investment, and a scope that is identical or better suited [35].

The gap between advanced systems with different characteristics against construction projects seems to impact the criteria for subcontractor selection. We found that system reliability and quality, level of service, and flexibility to changes are the highest-rate criteria and are highly relevant for management decisions in sophisticated technological industries. Given the dynamic environments, complexities, and uncertainties of the business world, reliability and quality considerations are essential for successful decision making in business scenarios. The requirements that characterize productive systems, i.e., high productivity, low price, and ability to compete with rapid changes are valid, regardless of sector and discipline [56]. Maintaining a technological system's functioning under continually changing conditions requires high reliability and ongoing improvement [57,58]. To reduce the probability of system failure, reliability in managerial decisions over the entire operational chain is essential, subcontractor selection included. In the defense industry, in particular, where system reliability is crucial, project managers and subcontractors must give the highest priority to reliability. Poor reliability can be costly and delay delivery, thus creating a shortage of spare parts and replacement systems and a rise in the price of the systems.

## 6. Summary

In determining the most dominant criteria for subcontractor selection in complex projects, understanding the direct impact of this choice on the quality of a project and its overall success is crucial. It is widely acknowledged that an external subcontractor must function as an integral part of the organization and provide a product or service whose quality is as high as that expected from an internal department. High-tech projects often encompass different fields that combine several engineering specializations with a leading core technology that is the expertise of the main contractor. The current study yielded several significant findings regarding the process of choosing a subcontractor for complex high-tech projects. The three significant highest-rated criteria were system reliability and quality, level of service, and flexibility to changes. The lowest-rated criteria were leadership, innovation, and number of systems supplied in the past.

These findings generate practical insights that help project managers select subcontractors for complex projects. With the evolution of high technologies, reliability-centered decisions concerning the value chain benefit the entire organization and significantly impact project success and overall outcomes.

*Study Limitations and Recommendations for Future Research*

The relatively small sample (*n* = 76) used in this study and the limited-diversity choice of high-tech companies restrict its broader applicability. Further research of more varied industries is required to broaden the findings' applicability.

Furthermore, a range of additional issues beyond the scope of the current study are worth exploring, e.g., the impact of system reliability and quality on the companies' strategic decisions, examining the effect of Industry 4.0-based technologies on subcontractor selection, and developing a model for choosing subcontractors in a dynamic, changing arena.

In order to increase the validity of the findings, additional research with a larger sample from diverse industries is necessary. Future research related to criteria for selecting subcontractors could use additional statistical methods, e.g., two-step cluster analysis, K-means cluster analysis, support vector machines, density-based clustering, Naïve Bayes classifier, or a classification tree model using the C5. algorithm. Moreover, the power of artificial intelligence (GPT- type) could be used to classify criteria based on an understanding of their textual content. Future studies might also analyze differences in the criteria of subcontractor selection for defense and civilian projects. It would also be interesting to examine the influence of project complexity on operational practices when selecting optimal subcontractors.

**Author Contributions:** Conceptualization, S.F. and S.K.; Data curation, S.F. and S.K.; Formal analysis, S.F. and S.K.; Investigation, S.F. and S.K.; Methodology, S.F. and S.K.; Resources, S.K.; Software, S.F.; Supervision, S.F. and S.K.; Validation, S.F. and S.K.; Writing—original draft, S.F. and S.K.; Writing—review and editing, S.K. All authors have read and agreed to the published version of the manuscript.

**Funding:** This research received no external funding.

**Institutional Review Board Statement:** The study was authorized by the institutional ethics committee of Ariel University, authorization number: AU-ENG-SK-20220818.

**Informed Consent Statement:** Informed consent was obtained from all subjects involved in the study.

**Conflicts of Interest:** The authors declare no conflict of interest.

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
