# Peer review of "Examining Criteria for Choosing Subcontractors for Complex and Multi-Systems Projects"

_sustainability, doi:10.3390/su142214988_

Round 1

Reviewer 1 Report

The paper entitled "Examining Criteria for Choosing Subcontractors for Complex 2 and Multi-Systems Projects". It talks about the process of choosing a subcontractor. 

Please consider the following methods.

- Please use more recent articles in the introduction section.

-Clearly note the advantages of the proposed method and list the innovative contributions.

-Discuss how to select the criteria. Is the questionnaire analyzed statistically? Please provide more about the statistics results.

- Please rewrite Figure 5. This table is not clear. Please use bigger-sized fonts for the results in this table.

-Is it possible to prove that the obtained measures in Figure 5 are unique? 

- Please note the limitations and further suggestions in the conclusion.

Author Response

Review 1 Report Form

We want to thank the reviewer for his comments. We found the diverse comments insightful, and we are convinced that they helped to improve the paper considerably. In this document, we detail how the comments were addressed in the revised paper.

As required, changes in the manuscript body are highlighted using the MS-Word Track Changes mechanism. There follows a description of the major changes in this version:

( )

( )

( )

( )

Comments and Suggestions for Authors

The paper entitled "Examining Criteria for Choosing Subcontractors for Complex 2 and Multi-Systems Projects". It talks about the process of choosing a subcontractor. 

Please consider the following methods.

- Please use more recent articles in the introduction section.

We added 6 recent articles to the manuscript.

-Clearly note the advantages of the proposed method and list the innovative contributions-

.

The following text was presented in the article in the introduction after the research question as a reference to the reviewer's comment:

In order to get an answer to the research question, the present study used one classical method (Friedman's Two-way Analysis of Variance by Ranks) and two additional advanced statistical methods: Hierarchical Cluster Analysis and Multidimensional Scaling (PROXSCAL). A combination of the three aforementioned methods is a necessary innovation in the current study. And this, for the following main reason: the sample in the current study was relatively small (convenience sample, 76 respondents) in relation to the 12 criteria. Therefore, the present study is at the seam between a quantitative approach and a qualitative approach. Friedman's Two-way Analysis of Variance by Ranks allows statistical inference from a sample to the population (ie, a quantitative approach) on the one hand. On the other hand, Hierarchical Cluster Analysis and Multidimensional Scaling (PROXSCAL) allows a qualitative approach, according to which the sample is viewed as a population, a kind of case study. In Hierarchical Cluster Analysis and Multidimensional Scaling (PROXSCAL) there are goodnesses of fits that indicate the uniqueness of the findings. Therefore, a combination of the above-mentioned three methods makes it possible to create an integrative view of the classification of the criteria for selecting subcontractors.

-Discuss how to select the criteria. Is the questionnaire analyzed statistically? Please provide more about the statistics results.

.

In stage 1 of the research, semi-structured interviews were conducted with six senior employees from academia and industry, as explained at the beginning of materials and methods section. The goal of the interviews was to identify the criteria used for choosing subcontractors for complex, multi-system projects.

The interviews were analyzed through qualitative content analysis (sorting the material into categories and themes and sorting the material within the themes) and by referring to the highest frequencies of the criteria specified by the interviewees. As a result, the 12 criteria were chosen to be presented in the questionnaire of stage 2.

- Please rewrite Figure 5. This table is not clear. Please use bigger-sized fonts for the results in this table.

FIGURE 5 has been fundamentally changed. Because Multidimensional Unfolding Analysis (PREFSCAL) had unconvincing visibility, Multidimensional Unfolding Analysis (PREFSCAL) was replaced by Multidimensional Scaling (PROXSCAL) which has a very strong visibility - and this is presented in the article. The accompanying tables were presented within textual information in the interpretation of the aforementioned chart. Below is FIGURE 5 (this is how it is presented in the article, FIGURE 5 in the article was changed to FIGURE 4):

It is important to note that FIGURE 5 appears [by mistake] in the article twice: And now the situation has been corrected: FIGURE 5 has become FIGURE 4, and FIGURE 6 has become FIGURE 5. FIGURE 5 (which was previously FIGURE 6) has also been significantly improved. Below is FIGURE 5 as it appears in the article:

-Is it possible to prove that the obtained measures in Figure 5 are unique?

The following new text is found in the article as a reference to the reviewer's above comment:

To further validate the above findings, Two-dimensional  Scaling Analysis (PROXSCAL) was used to reduce the dimensions of the 12 subcontractor selection criteria. The criteria were entered into the analysis as column objects and the 76 research participants were considered as row objects. Figure 4 shows the results..

Two-dimensional Scaling Analysis yielded excellent Stress and Fit Measures indicating a clear separation between groups of the criteria: Normalized Raw Stress = 0.00967, Stress-I = 0.098338 [Optimal scaling factor = 1.010], Stress-II = 0.177445 [Optimal scaling factor = 1.010], S-Stress = 0.013342 [Optimal scaling factor = 1.001], Dispersion Accounted For (D.A.F.) = 0.990330, Tucker's Coefficient of Congruence = 0.995153. From Figure 5 it can be clearly seen that Criterion 12: System reliability and quality, Criterion 06: Level of service, Criterion 03: Flexibility to changes are grouped together into a group (cluster) of criteria of very strong importance because their mean ranks are the highest and are in the range of 7.77- 8.46. In addition, it can be seen from Figure 5 that Criterion 01: Experience in the field, Criterion 09: Availability, Criterion 07: Quality standards & procedures are grouped together into a group (cluster) of criteria of strong importance because their mean ranks are in the range of 7.06 - 7.38. Figure 5 reveals another group of the following criteria that are relatively close to each other: Criterion 08: Complements missing abilities, Criterion 05: Pricing/costs, Criterion 11: Technological maturity, Criterion 04: Previous work relationships - these criteria are grouped together into a group (cluster ) of criteria of medium importance because their mean ranks are in the range of 5.79-5.52. And finally, Figure 5 reveals a final group of the following criteria which are also relatively close to each other: Criterion 02: Leadership and innovation, Criterion 10: Number of systems supplied in the past - these criteria are grouped together into a group (cluster) of criteria of weak importance because their mean ranks are in the range of 3.36-4.72.

That is, the results of the Two-dimensional Scaling Analysis (PROXSCAL) fit the results of the Related-samples Friedman's Two-way Analysis of Variance by Ranks including Pairwise Comparisons and the results of the Hierarchical Cluster Analysis.

Please note the limitations and further suggestions in the conclusion

We presented the limitations and further suggestions in the last part of the manuscript. The limitations were the small sample size and the diversity of industries that were included in the study. In order to increase the validity of the findings, additional research with a larger sample and various industries is necessary. Further studies might also analyze differences between the criteria for choosing subcontractors to work on defense and civilian projects. Also of interest would be examining the influence of project complexity on operational practices when selecting optimal subcontractors. There is a wide range of studies to be explored beyond the scope of the current study. For example, the impact of the system reliability and quality on the companies’ strategic decisions, examining the Industry 4.0-based technologies on the subcontractors' selection or developing a framework for choosing subcontractors in the arena of dynamic changing.    

The following new text is in the last paragraph of the article:

In the future research that will be related to the selection of the criteria, it is possible to harness additional statistical methods such as TwoStep Cluster Analysis, K-Means Cluster Analysis, Support Vector Machines, Density-Based Clustering, Naïve Bayes Classifier, classification tree model using the C5.0 algorithm. Also, it will be possible to harness the power of artificial intelligence of the GPT-3 type, which will try to classify criteria based on an understanding of the textual content of these criteria.

  •  

Reviewer 2 Report

The study is empirical and is devoted to the complex problem of choosing the optimal solution under uncertainty. Undoubtedly, such a decision is influenced by the criteria by which the choice is made. Therefore, the authors focused their attention on evaluating the criteria and determining their relationships. The proposed solution methods are relevant, but they are not sufficiently described in the article, as well as the initial data for the study. In general, the paper is consistent and logical, and it is illustrated and confirmed by relevant calculations and diagrams. The inscriptions in figure 2 and figure 5 are not very clear and in small print, which can cause difficulty in reading for people with poor eyesight. There are very few modern articles in the list of references, which does not give an idea of ​​the current state of this issue. The article has potential for improvement. The authors should more clearly formulate the purpose and objectives of the study, as well as conclusions, separating them from the discussion.

Author Response

Review 2 Report Form

We want to thank the reviewer for his comments. We found the diverse comments insightful, and we are convinced

that they helped to improve the paper considerably. In this document, we detail how the comments were

addressed in the revised paper.

As required, changes in the manuscript body are highlighted using the MS-Word Track Changes mechanism.

There follows a description of the major changes in this version:

Comments and Suggestions for Authors

The study is empirical and is devoted to the complex problem of choosing the optimal solution under uncertainty. Undoubtedly, such a decision is influenced by the criteria by which the choice is made. Therefore, the authors focused their attention on evaluating the criteria and determining their relationships. The proposed solution methods are relevant, but they are not sufficiently described in the article, as well as the initial data for the study. In general, the paper is consistent and logical, and it is illustrated and confirmed by relevant calculations and diagrams. The inscriptions in figure 2 and figure 5 are not very clear and in small print, which can cause difficulty in reading for people with poor eyesight. There are very few modern articles in the list of references, which does not give an idea of ​​the current state of this issue. The article has potential for improvement. The authors should more clearly formulate the purpose and objectives of the study, as well as conclusions, separating them from the discussion.

Our answer to reviewer 2:

We added some modern references to the literature review and the discussion. The references relate to the main topics of the manuscript:  Choosing Subcontractors and Complex and Multi-Systems Projects.

The purpose and objectives of the study are redefined in the introduction section.

We separated the conclusions from the discussion. We also enlarged both sections and added some updated references for discussion. 

FIGURE 2 and FIGURE 5 (in the article FIGURE 5 is FIGURE 4) have been significantly changed. The following new text is found as accompanying text for FIGURE 2 in the article:

The findings of the analysis imply that the 12 subcontractor selection criteria can be sorted into four groups (clusters) that differ from each other in the strength. Figure 2 shows a possible division of the 12 criteria into four groups (clusters).

The following new text is found as accompanying text for FIGURE 4 in the article:

To further validate the above findings, Two-dimensional  Scaling Analysis (PROXSCAL) was used to reduce the dimensions of the 12 subcontractor selection criteria. The criteria were entered into the analysis as column objects and the 76 research participants were considered as row objects. Figure 4 shows the results..

Two-dimensional Scaling Analysis yielded excellent Stress and Fit Measures indicating a clear separation between groups of the criteria: Normalized Raw Stress = 0.00967, Stress-I = 0.098338 [Optimal scaling factor = 1.010], Stress-II = 0.177445 [Optimal scaling factor = 1.010], S-Stress = 0.013342 [Optimal scaling factor = 1.001], Dispersion Accounted For (D.A.F.) = 0.990330, Tucker's Coefficient of Congruence = 0.995153. From Figure 5 it can be clearly seen that Criterion 12: System reliability and quality, Criterion 06: Level of service, Criterion 03: Flexibility to changes are grouped together into a group (cluster) of criteria of very strong importance because their mean ranks are the highest and are in the range of 7.77- 8.46. In addition, it can be seen from Figure 5 that Criterion 01: Experience in the field, Criterion 09: Availability, Criterion 07: Quality standards & procedures are grouped together into a group (cluster) of criteria of strong importance because their mean ranks are in the range of 7.06 - 7.38. Figure 5 reveals another group of the following criteria that are relatively close to each other: Criterion 08: Complements missing abilities, Criterion 05: Pricing/costs, Criterion 11: Technological maturity, Criterion 04: Previous work relationships - these criteria are grouped together into a group (cluster ) of criteria of medium importance because their mean ranks are in the range of 5.79-5.52. And finally, Figure 5 reveals a final group of the following criteria which are also relatively close to each other: Criterion 02: Leadership and innovation, Criterion 10: Number of systems supplied in the past - these criteria are grouped together into a group (cluster) of criteria of weak importance because their mean ranks are in the range of 3.36-4.72.

Reviewer 3 Report

This paper is about Criteria for Choosing Subcontractors for Complex and Multi-Systems Projects, an exciting research area. There are many application areas of this research, like MCDM. It is well-written and sound-structured. However, there are two bottlenecks in this paper.

1) 12 criteria were selected by the author. The scope of this research should be improved. At least 25 or 30 criteria could significantly improve the quality of this research.

2) The number of samples is insufficient for the journal's quality. The author has to improve the number of samples. The general approach is the number of questions x 10 samples needed.

Author Response

Review 3 Report Form

We want to thank the reviewer for his comments. We found the diverse comments insightful, and we are convinced that they helped to improve the paper considerably. In this document, we detail how the comments were addressed in the revised paper.

As required, changes in the manuscript body are highlighted using the MS-Word Track Changes mechanism. There follows a description of the major changes in this version:

Comments and Suggestions for Authors

This paper is about Criteria for Choosing Subcontractors for Complex and Multi-Systems Projects, an exciting research area. There are many application areas of this research, like MCDM. It is well-written and sound-structured. However, there are two bottlenecks in this paper.

1) 12 criteria were selected by the author. The scope of this research should be improved. At least 25 or 30 criteria could significantly improve the quality of this research

Thank you for the significant comment. The process of selecting the 12 criteria is explained at the beginning of materials and methods section. In stage 1 of the research, semi-structured interviews were conducted with six senior employees from academia and industry. The goal of the interviews was to identify the criteria used for choosing subcontractors for complex, multi-system projects. The interviews were analyzed to find common criteria among the interviewees for choosing sub-contractors.

The interviews were analyzed through qualitative content analysis (sorting the material into categories and themes and sorting the material within the themes) and by referring to the highest frequencies of the criteria specified by the interviewees. As a result, the 12 criteria were chosen to be presented in the questionnaire of stage 2.

2) The number of samples is insufficient for the journal's quality. The author has to improve the number of samples. The general approach is the number of questions x 10 samples needed.

In order to get an answer to the research question, the present study used one classical method (Friedman's Two-way Analysis of Variance by Ranks) and two additional advanced statistical methods: Hierarchical Cluster Analysis and Multidimensional Scaling (PROXSCAL). A combination of the three aforementioned methods is a necessary innovation in the current study. And this, for the following main reason: the sample in the current study was relatively small (convenience sample, about 76 respondents) in relation to the 12 criteria. Therefore, the present study is at the seam between a quantitative approach and a qualitative approach. Friedman's Two-way Analysis of Variance by Ranks allows statistical inference from a sample to the population (ie, a quantitative approach) on the one hand. On the other hand, Hierarchical Cluster Analysis and Multidimensional Scaling (PROXSCAL) allow a qualitative approach, according to which the sample is viewed as a population, a kind of case study. In Hierarchical Cluster Analysis and Multidimensional Scaling (PROXSCAL) there are goodnesses of fits that indicate the uniqueness of the findings. Therefore, a combination of the above-mentioned three methods makes it possible to create an integrative view of the classification of the criteria for selecting subcontractors.

Reviewer 4 Report

Abstract:

·       The author use capital letters in some words. This is not accepted.

·       The scope, the methodology, the utilized analytical techniques are not clear in the abstract and fuzziness.

·       Please do not use pronouns when writing.

Introduction:

·       This section is poorly written. It doesn't clearly define the scope of the study, the gap that it will address, contribution, and why it is important. Further, the writing and language have many mistakes.

Literature Review:

·       I can't understand what the author wants to inform from this review. It includes mixed data without any arrangement. It doesn't reflect the prior works attempts or their gaps. The flow of the sentences and paragraphs is very hard for being understood. This section must be rewritten and restructured. Further, many points have been introduced without references. 

Materials and Methods:

Please bridge the following gaps in the methodology:

1.    How the factors have been identified from literature?

2.    Population of the study.

3.    How the sample size has been determined and designed?

4.    What about the sample of the pilot study?

5.    How the validity and reliability of the questionnaire have been achieved?

6.    How the experts have been selected and surveid?

7.    How the questionnaires have been sent and received from the experts?

8.    What about the realized sample size and response rate and its appetence?

9.    What about the validity and reliability of the collected data?

As the previous sections in the manuscript the flow and the structure of the presented paragraphs are very poor. Further, the author lists the utilized analytical techniques without any descriptions or justifications.

Results: 

·       The figures, tables, and content of this section are not clear and fuzziness. Really, I struggle to understand how the results have been obtained relying upon the used analytical techniques. Further, the presentation of the results is very hard for being understood.

Discussion and Conclusions

·       The author list some of the findings and some of the scope of the study without any discussion. The implications and limitations are not mentioned clearly and need to be extended.   

Author Response

Review 4- Report Form

We want to thank the reviewer for his comments. We found the diverse comments insightful, and we are convinced that they helped to improve the paper considerably. In this document, we detail how the comments were addressed in the revised paper.

As required, changes in the manuscript body are highlighted using the MS-Word Track Changes mechanism. There follows a description of the major changes in this version:

Comments and Suggestions for Authors

Abstract:

  • The author use capital letters in some words. This is not accepted

We changed the use of capital letters.

  • The scope, the methodology, the utilized analytical techniques are not clear in the abstract and fuzziness.

We rewrote the relevant sections in the paper. We also used a linguistic and academic editor to improve the wording in the manuscript.  

  • Please do not use pronouns when writing.

We changed this accordingly.

Introduction:

  • This section is poorly written. It doesn't clearly define the scope of the study, the gap that it will address, contribution, and why it is important. Further, the writing and language have many mistakes.

We rewrote the introduction. We also

Literature Review:

  • I can't understand what the author wants to inform from this review. It includes mixed data without any arrangement. It doesn't reflect the prior works attempts or their gaps. The flow of the sentences and paragraphs is very hard for being understood. This section must be rewritten and restructured. Further, many points have been introduced without references

Our answer

We rearranged the literature review. In the first part of the literature review, we cited the definition of subcontracting according to the PMI. Then we presented references related to the selection and management of subcontractors and the criteria for selecting subcontractors (Mainly in the construction industry).

In the second part of the literature review, we presented references to complex projects since the study examined such projects. We cited a complex project definition according to some references and the project complexity through organizational, cultural, environmental, technological, and information complexity lenses.

We added some modern references to the literature review and the discussion. The references relate to the main topics of the manuscript:  Choosing Subcontractors and Complex and Multi-Systems Projects.

Materials and Methods:

Please bridge the following gaps in the methodology:

  1. How the factors have been identified from literature

We expect that the degree of importance of the criteria for selecting subcontractors will be spread over a certain spectrum that includes at one end of the spectrum a collection of criteria with the highest degree of importance and at the other end of the spectrum the collection of criteria with the lowest degree of importance. The research design in the present study was designed to discover this.

  1. Population of the study

The population included professional employees with recognized competency in engineering and management from the civilian and defense industries. The various industries, from large multinational companies to small companies, provide and develop a specific technology. The sample was taken from the professional employees included 76 respondents who have been involved in projects of different sizes: 45 respondents from the civilian sector and 31 from defense industries. The positions represented included systems engineer, developer, purchasing manager and operations manager. All respondents were directly involved in and/or affiliated with choosing subcontractors.

  1. How the sample size has been determined and designed?

The sampling was conducted according to convenience considerations (convenience sampling). Convenience sampling is the inclusion in a sample of people who are easy to reach for the purpose of the study. These are accessible and available people, who are willing to volunteer to be included in the research or who can be relatively easily recruited to participate in the research. It should be emphasized that in this type of sampling not every unit in the population has an equal opportunity to be included in the sample. Also, in this process of sampling, it is not possible to know in advance the characteristics of the sampled and their suitability to provide reliable information as required in the study in question.  Therefore, we happily settled for 76 respondents.

  1. What about the sample of the pilot study?

We presented the information about the pilot study at the first paragraph of the materials and methods section:

In the pilot study , semi-structured interviews were conducted with six senior expert employees from academia and industry. All of the interviewees had a rich background and broad experience in engineering systems, purchasing management and working with subcontractors.

  1. How the validity and reliability of the questionnaire have been achieved?

The questionnaire’s validity was checked by five experts. The first version of the questionnaire was distributed to those experts who were asked to assess the extent of relevancy and clarity of each questionnaire item in a scale from 1 (very low) to 5 (very high), and to suggest changes and/or corrections.

After receiving this feedback from the experts, we used Fleiss Kappa procedure to evaluate the degree of agreement among the experts regarding to the relevancy and clarity of the questionnaire items. Fleiss Kappa index amounted to 0.78, which indicates substantial agreement between the experts. In addition, several changes and updates were made in the wording of the questionnaire.

Reliability was examined by checking for internal consistency using the questionnaire’s Alpha (Cronbach) reliability. The result was 0.829, which reflects the questionnaire’s high internal reliability.

  1. How the experts have been selected and surveid?

The experts represented the “state of the art” in their discipline: systems engineering, project management, complex projects, and sub-contractors in defense and civilian industries. They were selected according to their high-quality professionality.

  1. How the questionnaires have been sent and received from the experts?

The first version of the questionnaire was distributed to the experts who were asked to assess the extent of relevancy and clarity of each questionnaire item in a scale from 1 (very low) to 5 (very high), and to suggest changes and/or corrections. We sent the questionnaires via google forms and received the experts' feedback on the same platform.

  1. What about the realized sample size and response rate and its appetence?

Because it is a convenience sample - the questionnaire was submitted to 76 respondents and all 76 respondents answered the questionnaire.

  1. What about the validity and reliability of the collected data?

The questionnaire’s validity was checked by five experts. The experts represented the “state of the art” in their discipline: systems engineering, project management, complex projects, and sub-contractors in defense and civilian industries. The first version of the questionnaire was distributed to those experts who were asked to assess the extent of relevancy and clarity of each questionnaire item in a scale from 1 (very low) to 5 (very high), and to suggest changes and/or corrections.

After receiving this feedback from the experts, we used Fleiss Kappa procedure to evaluate the degree of agreement among the experts regarding to the relevancy and clarity of the questionnaire items. Fleiss Kappa index amounted to 0.78, which indicates substantial agreement between the experts. In addition, several changes and updates were made in the wording of the questionnaire.

Reliability was examined by checking for internal consistency using the questionnaire’s Alpha (Cronbach) reliability. The result was 0.83, which reflects the questionnaire’s high internal reliability.

As the previous sections in the manuscript the flow and the structure of the presented paragraphs are very poor. Further, the author lists the utilized analytical techniques without any descriptions or justifications.

About the validity and about " As the previous sections in the manuscript the flow and the structure of the presented paragraphs are very poor. Further, the author lists the utilized analytical techniques without any descriptions or justifications.":

Responses to the questionnaire was analyzed using IBM SPSS 27.0 [49]. Before testing the research questions, the distribution of 12 selection criteria was checked. Next, to answer the research question, a Related-samples Friedman’s Two-way Analysis of Variance by Ranks was conducted, including Pairwise Comparisons Analysis. The Friedman Test, as a nonparametric test, is used with ranked data, particularly for when the data do not meet the rigor of interval data. An addition, there are serious concerns about extreme deviation from normal distribution. Friedman’s test doesn’t assume data comes from a particular distribution (like the normal distribution). Basically, it’s used in place of the ANOVA test when don’t know the distribution of data. Friedman’s test is an extension of the sign test, used when there are multiple treatments. In fact, if there are only two treatments the two tests are identical. Those findings were then validated using Hierarchical Cluster Analysis. Hierarchical Cluster Analysis attempts to identify relatively homogeneous groups of cases (or variables) based on selected characteristics, using an algorithm that starts with each case (or variable) in a separate cluster and combines clusters until only one is left. It is possible to analyze raw variables, or it is possible to choose from a variety of standardizing transformations. Distance or similarity measures are generated by the Proximities procedure. Statistics are displayed at each stage to help select the best solution. Finally, Multidimensional Scaling (PROXSCAL) was used to further validate the findings of the first two statistical tests. Multidimensional Scaling (PROXSCAL) attempts to find the structure in a set of proximity measures between objects. This process is accomplished by assigning observations to specific locations in a conceptual low-dimensional space such that the distances between points in the space match the given (dis)similarities as closely as possible. The result is a least-squares representation of the objects in that low-dimensional space, which, in many cases, will help to further understand data.

The results of the Two-dimensional Scaling Analysis (PROXSCAL) fit the results of the Related-samples Friedman's Two-way Analysis of Variance by Ranks including Pairwise Comparisons and the results of the Hierarchical Cluster Analysis.

Results: 

  • The figures, tables, and content of this section are not clear and fuzziness. Really, I struggle to understand how the results have been obtained relying upon the used analytical techniques. Further, the presentation of the results is very hard for being understood.

We rewrote the results section

Discussion and Conclusions

  • The author list some of the findings and some of the scope of the study without any discussion. The implications and limitations are not mentioned clearly and need to be extended.   

We separated the conclusions from the discussion. We also enlarged both sections and added some updated references for discussion. 

We presented the limitations and further suggestions in the last section of the manuscript.

It appears in the article like this:

Study Limitations and Recommendations for Future Research

The broader applicability of this study might be limited by its relatively small sample (N=76). In addition, the diversity of the industries was limited and concentrated on high-tech companies.  In order to increase the validity of the findings, additional research with a larger sample and various industries is necessary. In the future research that will be related to the selection of the criteria, it is possible to harness additional statistical methods such as TwoStep Cluster Analysis, K-Means Cluster Analysis, Support Vector Machines, Density-Based Clustering, Naïve Bayes Classifier, classification tree model using the C5.0 algorithm. Also, it will be possible to harness the power of artificial intelligence of the GPT-3 type, which will try to classify criteria based on an understanding of the textual content of these criteria. Further studies might also analyze differences between the criteria for choosing subcontractors to work on defense and civilian projects. Also of interest would be examining the influence of project complexity on operational practices when selecting optimal subcontractors. There is a wide range of studies to be explored beyond the scope of the current study. For example, the impact of the system reliability and quality on the companies’ strategic decisions, examining the Industry 4.0-based technologies on the subcontractors' selection or developing a framework for choosing subcontractors in the arena of dynamic changing.    

Round 2

Reviewer 1 Report

This version of the manuscript is now much better.

But, please put more time into the following comments.

1- Please revise the introduction in referring to more related existing papers for literature review.

2-Mentioning more adequate articles for focusing on the advantages and disadvantages of the existing models compared to the newly presented one in this paper.

3-Also, please clearly discuss the uniqueness of ranks.

Author Response

Response to round 2 comments on the paper:

As before, we want to thank the reviewer again for his comments, helping us to improve the paper considerably. In this document, we detail how the comments were addressed in the revised paper.

As required, changes in the manuscript body are highlighted using the MS-Word Track Changes mechanism. There follows a description of the major changes in this version:

1- Please revise the introduction in referring to more related existing papers for literature review.

On p. 2 we added source

15 (Vo, K.D., Pham, C.P., Phan, P.T., Vu, N.B., Duong, M.T.H., Le, L.P., Nguyen, Q.L.H.T.T. (2021). Critical factors of subcontractor evaluation and selection: A case study in Vietnam. J. Asian Finance Econ. Bus. 2021, 8(3), 297–305. https://doi.org/10.13106/jafeb.2021.vol8.no3.0297)

and source 55

(Koçak, S., Kazaz, A., Ulubeyli, S. Subcontractor selection with additive ratio assessment method. J. Constr. Eng. Manag. Innov. 2018, 1(1), 18–32. https:// doi.org /10.31462/jcemi.2018.01018032)

into the following paragraph:

A review of the academic literature shows reveals that only a few studies analyze the selection of subcontractors or develop a model framework for the selection process. Most of the prior studies concentrate on engagement with the construction community companies [2,3, 15, 55]. However, the criteria for subcontractor selection for other domains, has not yet been elucidated, so it is required to address the issue to further studies.

2-Mentioning more adequate articles for focusing on the advantages and disadvantages of the existing models compared to the newly presented one in this paper.

On page 2, at the end of the chapter "Research Question" we added the following text:

The combination of the three aforementioned methods is an advantage over other statistical methods, such as a fuzzy analytic network process analysis [43] as long as these methods are produced only and there is no simultaneous validation using other static methods based on a different mathematical basis in the same study.

43: He, Q.H.; Luo, L., Hu, Y.; Chan, A.P.C. Measuring the complexity of mega construction projects in China-A fuzzy analytic network process analysis, Int. J. Proj. Manag. 2015, 33(3), 549–563. https://doi.org/10.1016/j.ijproman.2014.07.009.

3-Also, please clearly discuss the uniqueness of ranks.

At the beginning of the chapter "Statistical analyses" (on page 7) we added the following text:

The statistical analysis in the current study was based on three nonparametric statistical methods: a related-samples Friedman's Two-way Analysis of Variance by Ranks, Hierarchical Cluster Analysis and Multidimensional Scaling (PROXSCAL). The uniqueness of ranks is reflected in the fact that a rank variable represents the ordering of the values of a numeric variable. Because ranks are the cornerstone of many non-parametric statistical methods, it is useful to compute the rank transform of a variable in a dataset.

Please revise the previous comments and do them "The figures, tables, and content of this section are not clear and fuzziness. Really, I struggle to understand how the results have been obtained relying upon the used analytical techniques. Further, the presentation of the results is very hard for being understood".

Shimon:

The research consisted of several stages. The first step is a necessary step and it proves non-normality of the distribution of the respondents' answers in the study. From here, the following steps follow: step 2, step 3 and step 4. Stages 2, 3 and 4 are the stages that can be independent stages, that is, it is possible to present in the current study only stage 1 + stage 2 only, or only stage 1 + stage 3 only, or only stage 1 + stage 4 only. The findings that yield stage 1+ and only stage 2 only, and stage 1 + and only stage 3 only, and stage 1 + and only stage 4 lead to the same conclusion that there are 4 groups of criteria that are on a spectrum, where at one end is a group of the weakest criteria and at the other end of the spectrum is a group of criteria with the highest intensity. But the ultimate challenge in the research was to combine steps 1-4 together to validate the truth of the spectrum. And this is what was done in the present study. After the representation of a non-normal distribution of the distribution of the respondents' answers, and after the reasoning of using ranking variables, a Related-samples Friedman's Two-way Analysis of Variance by Ranks test was conducted.

To validate the insights obtained from Friedman’s Two-way Analysis, a Hierarchical Cluster Analysis was conducted, in which the 12 subcontractor selection criteria were used as cases, and the mean rank was used as a classification variable. A Hierarchical Cluster Analysis persistently repeated the same structure of the groups:

But it was not enough because both methods could have been equally misleading. Therefore, the 3rd method was produced - two-dimensional PROXSCAL analysis. If the 3rd method would yield the same insights then it would be argued that there are 4 groups of criteria. And this is what happened: the third method with excellent model quality indicators yielded (on the basis of visualization of proximity between the criteria) again a division into 4 groups of the criteria:

Thus, the results of the two-dimensional scaling analysis (PROXSCAL) fit the results of Friedman's Two-way Analysis and the Hierarchical Cluster Analysis.

Reviewer 3 Report

The author made the necessary explanations and revisions. The revised paper could publish as it is.

Author Response

Thank you very much for your feedback.

I appreciate your contribution to improving the paper.  

Reviewer 4 Report

Very little efforts have been conducted to improve the paper. Thus, the next comments need to be addressed.  

Abstract:

·       This section now needs further improvement to clearly present the scope and the methodology of the paper. Additionally, the paper should be reviewed by an English editor to improve its language. 

Introduction:

·       The gap is not strong to support the importance of the study.

·       Please do not use the pronouns.

·       The flow of the data of the whole section and its subsection is very bad. The authors gather some sentences together without any meaning.

Literature Review:

·       The authors list the role of the subcontractor or associated definitions. The authors should list the prior works, their gaps. Hence, their contributions to the current study can appear.

·       Many paragraphs have been introduced without references.  

Materials and Methods:

·       I cant understand whether the interviews were conducted to extract the criteria from the experts or to validate the criteria. How the interviews have been conducted without listing anything about the criteria.

·       The authors target 76 engineers and all of them answered the questionnaires. Why this number is selected.

·       I can not imagine how the questionnaire was distributed to 76 engineers and then its validity has been examined by 5 leaders.

·       The sequence of the methodology has fatal mistakes. Please revise the previous comments and do them.  

Results: 

·       Please revise the previous comments and do them "The figures, tables, and content of this section are not clear and fuzziness. Really, I struggle to understand how the results have been obtained relying upon the used analytical techniques. Further, the presentation of the results is very hard for being understood".

Discussion and Conclusions

·       The authors summarize their work and some improvements have been realized. However, more in depth discussions are highly needed. The implications are not mentioned clearly and need to be extended.   

Author Response

Response to round 2 comments on the paper:

We thank the reviewer again for the enlightening comments, which helped us improve the paper. Below, we describe how we addressed the comments in the revised version.

Per your instruction, we used the MS Word Track Changes mechanism to highlight the changes made to the manuscript body . Following is a summary description of the main changes made in this version:

Abstract:

  • This section now needs further improvement to clearly present the scope and the methodology of the paper. Additionally, the paper should be reviewed by an English editor to improve its language. 

We revised the abstract and added to it the methodology of the study.

The paper was re-edited, and some improvements have been entered.

Introduction:

  • The gap is not strong to support the importance of the study.
  • Please do not use the pronouns.
  • The flow of the data of the whole section and its subsection is very bad. The authors gather some sentences together without any meaning.

We rewrote the introduction and clarified the gap to support the importance of the study.

Literature Review:

  • The authors list the role of the subcontractor or associated definitions. The authors should list the prior works, their gaps. Hence, their contributions to the current study can appear.
  • Many paragraphs have been introduced without references.  

We revised the literature review and added the missing references.

Materials and Methods:

  • I cant understand whether the interviews were conducted to extract the criteria from the experts or to validate the criteria. How the interviews have been conducted without listing anything about the criteria.
  • The authors target 76 engineers and all of them answered the questionnaires. Why this number is selected.
  • I can not imagine how the questionnaire was distributed to 76 engineers and then its validity has been examined by 5 leaders.
  • The sequence of the methodology has fatal mistakes. Please revise the previous comments and do them

We clarified the order of the study's methodical stages and explained how the questionnaire was put together and validated before it was sent to the respondents. We also added a figure of the study design.

Results

Please revise the previous comments and do them "The figures, tables, and content of this section are not clear and fuzziness. Really, I struggle to understand how the results have been obtained relying upon the used analytical techniques. Further, the presentation of the results is very hard for being understood".

We revised the results section according to this comment as follows:

The research consisted of several stages. The first step is a necessary step and it proves non-normality of the distribution of the respondents' answers in the study. From here, the following steps follow: step 2, step 3 and step 4. Stages 2, 3 and 4 are the stages that can be independent stages, that is, it is possible to present in the current study only stage 1 + stage 2 only, or only stage 1 + stage 3 only, or only stage 1 + stage 4 only. The findings that yield stage 1+ and only stage 2 only, and stage 1 + and only stage 3 only, and stage 1 + and only stage 4 lead to the same conclusion that there are 4 groups of criteria that are on a spectrum, where at one end is a group of the weakest criteria and at the other end of the spectrum is a group of criteria with the highest intensity. But the ultimate challenge in the research was to combine steps 1-4 together to validate the truth of the spectrum. And this is what was done in the present study. After the representation of a non-normal distribution of the distribution of the respondents' answers, and after the reasoning of using ranking variables, a Related-samples Friedman's Two-way Analysis of Variance by Ranks test was conducted. The analysis suggested the following division:

To validate the insights obtained from Friedman's Two-way Analysis, a Hierarchical Cluster Analysis was conducted, in which the 12 subcontractor selection criteria were used as cases, and the mean rank was used as a classification variable. A Hierarchical Cluster Analysis persistently repeated the same structure of the groups:

But it was not enough because both methods could have been equally misleading. Therefore, the 3rd method was produced - two-dimensional PROXSCAL analysis. If the 3rd method would yield the same insights then it would be argued that there are 4 groups of criteria. And this is what happened: the third method with excellent model quality indicators yielded (on the basis of visualization of proximity between the criteria) again a division into 4 groups of the criteria:

Thus, the results of the two-dimensional scaling analysis (PROXSCAL) fit the results of Friedman's Two-way Analysis and the Hierarchical Cluster Analysis.

Discussion and Conclusions

  • The authors summarize their work and some improvements have been realized. However, more in depth discussions are highly needed. The implications are not mentioned clearly and need to be extended.

We revised this section and added the study's implications. We also dwelt on the significance of the results and the importance of the high-rated criteria for selecting subcontractors.    

Round 3

Reviewer 4 Report

The whole language of the paper has not been improved. For example, see the first two lines in the abstract. 

The structure of the section and sub-sections of the paper are very bad. For example, the introduction includes two paragraphs, then research goals, research question. Further, in the research question the analysis tools.

In the Literature review some words have been changed and the same issue as has been mentioned in the prior rounds are still existed.

The methodology has been slightly improved.

The results are as the previous versions without any improvements.

Author Response

Dear reviewer,

Thank you for your comments.

Further to your comments, we made the following additional emendations:

The whole language of the paper has not been improved. For example, see the first two lines in the abstract. 

We rewrote the two lines.

The structure of the section and sub-sections of the paper are very bad. For example, the introduction includes two paragraphs, then research goals, research question. Further, in the research question the analysis tools.

We combined the research goals and research questions in the introduction section.The introduction now includes an overview of the subject, and a closing paragraph that briefly describes the study's purpose and research question. The analysis tools were moved to the Methodology section.

In the Literature review, some words have been changed and the same issue as has been mentioned in the prior rounds are still existed.

The literature review overviews the current state of research in the study's field of interest and covers its two main topics: Subcontractors and

The methodology has been slightly improved.

The results are as the previous versions without any improvements.

The results section represents the core findings of our study derived from the statistical methods applied to analyze the questionnaire data. The findings are presented in a logical sequence and answer our study questions. We have made several modifications in this section following the reviewer's comments. An expert in advanced statistics did all the statistical analysis.
